# Adoption of the 2A Ribosomal Skip Principle to Track Assembled Virions of Pepper Mild Mottle Virus in *Nicotiana benthamiana*

**DOI:** 10.3390/plants13070928

**Published:** 2024-03-22

**Authors:** Mengting Jiao, Yueyan Yin, Yanzhen Tian, Jianing Lei, Lin Lin, Jian Wu, Yuwen Lu, Hongying Zheng, Fei Yan, Jianguang Wang, Jiejun Peng

**Affiliations:** 1State Key Laboratory for Managing Biotic and Chemical Threats to the Quality and Safety of Agroproducts, Institute of Plant Virology, Ningbo University, Ningbo 315211, China; mengtingjiao96@163.com (M.J.); tianyanzhen@nbu.edu.cn (Y.T.); ljn13849163355@163.com (J.L.); linlin1@nbu.edu.cn (L.L.); wujian@nbu.edu.cn (J.W.); luyuwen@nbu.edu.cn (Y.L.); zhenghongying@nbu.edu.cn (H.Z.); yanfei@nbu.edu.cn (F.Y.); 2Key Laboratory of Biotechnology in Plant Protection of MARA and Zhejiang Province, Institute of Plant Virology, Ningbo University, Ningbo 315211, China; 3Biocontrol Engineering Research Center of Crop Disease & Pest of Yunnan Province, School of Ecology and Environmental Science, Yunnan University, Kunming 650091, China; 4Institute of Biotechnology and Germplasm Resources, Yunnan Academy of Agricultural Sciences, Kunming 650223, China; yinyy0324@163.com

**Keywords:** 2A, pepper mild mottle virus, coat protein, self-assembly

## Abstract

The coat protein (CP) is an important structural protein that plays many functional roles during the viral cycle. In this study, the CP of pepper mild mottle virus (PMMoV) was genetically fused to GFP using the foot-and-mouth disease virus peptide 2A linker peptide and the construct (PMMoV-GFP^2A^) was shown to be infectious. The systemic spread of the virus was monitored by its fluorescence in infected plants. Electron microscopy and immunocolloidal gold labelling confirmed that PMMoV-GFP^2A^ forms rod-shaped particles on which GFP is displayed. Studies of tissue ultrastructure and virion self-assembly confirmed that PMMoV-GFP^2A^ could be used to monitor the real-time dynamic changes of CP location during virus infection. Aggregations of GFP-tagged virions appeared as fluorescent plaques in confocal laser microscopy. Altogether, PMMoV-GFP^2A^ is a useful tool for studying the spatial and temporal changes of PMMoV CP during viral infection.

## 1. Introduction

Virus coat protein (CP) is an important structural protein that is crucial for establishing infection [1,2]. The CPs of plant viruses play a key role in protecting the viral genome and also participate in viral cell-to-cell and systemic movement. The free CPs interact with other viral/host proteins to form replication complexes, modulate host defense pathways, and process host mRNA [3,4,5]. Viruses classified in the genus *Tobamovirus* (family *Virgaviridae*) include very harmful pathogens of the most important crops [6]. Tobamoviruses have a positive-strand RNA genome that encodes four proteins. The replication-associated proteins p126 and p183 (RNA-dependent RNA polymerase, RdRp, produced by translational readthrough) are translated from the full-length RNA while the MP (movement protein) and CP are translated from separate subgenomic RNAs [6,7]. The tobamovirus CP has many functions during the viral cycle [3,8] and is known to be involved in encapsidation, systemic movement, pathogenicity, and symptom development [9,10,11]. The rod-shaped virus particle, approximately 300 nm in length and 18 nm in diameter, contains about 2130 copies of the CP.

Pepper mild mottle virus (PMMoV) is an important tobamovirus that is a major viral pathogen of solanaceous plants and is universally present in sauce, human feces, water, soil, etc. [12,13,14]. Previous studies have shown that substitutions at amino acids 43 and 50 of PMMoV CP enable the virus to overcome the L3 resistance, derived from *Capsicum frutescens* [15]. A single amino acid in the CP determines its subcellular localization and the chlorotic symptoms on the leaves of pepper [10]. Recent work also shows that the PMMoV CP interacts with pepper chloroplast outer envelope membrane protein (OMP24) to impair OMP24-induced stromules, perinuclear chloroplast clustering, and ROS accumulation [16]. These findings suggest that PMMoV CP can modulate the plant defense to facilitate viral infection. While the role of the CP is significant, little is known of the spatial and temporal changes of the CP during the viral cycle. A study of changes in the distribution of tobacco mosaic virus (TMV) CP in infected protoplasts using immunofluorescence and 3D microscopy have shown that the CP enhances the production of MP and increases the size of the VRC (Virus Replication Complex) [17]. However, there have been no studies of changes in the CP distribution (especially virion assembly) within plants during the viral cycle and particularly in systemic (non-inoculated) leaves.

In vivo, noninvasive virus labeling and virus tracking are effective tools for studying the viral cycle. A common method of studying the systemic movement of tobamoviruses involves translating the green fluorescent protein (GFP) from the CP subgenomic mRNA promoter [18,19,20]. This results in the expression of free GFP in cells where virus protein is being produced. There is rapidly growing interest in using plant viral nanoparticles for the packaging and in vivo delivery of bioactive cargos [21]. Two strategies have been used to display cargo (foreign protein or peptides) from tobamoviruses [22,23,24]. In the first, the cargo protein is displayed on the surface of a tobamovirus as a C-terminal fusion to the coat protein via a 15-aa linker (flexible linkers, (GGGGS)_n_) [24,25,26]. An alternative method is to insert a foot-and-mouth disease virus 2A sequence between the cargo protein and the CP [22,23]. The 2A (derived from foot-and-mouth disease virus) is a mere 18 amino acids long and serves as a bond skipping mechanism between the carboxy-terminal glycine residue of the peptide, based on the ribosomal skip principle (forming cargo-2A-CP, free cargo, and free CP) [27,28]. The cargo protein fused with the 2A peptide linker can be formed into tobamovirus virions that move systemically [22,23]. However, the real-time dynamics of CPs or virions have not been clearly described in infected cells of systemic leaves.

The PMMoV-based vector system for a high-level production of heterologous protein in plants has been reported. However, the process of constructing a chimeric virus for studying the systemic movement or labeling the virion of PMMoV has not been addressed [19]. In this study, we genetically fused the GFP to the PMMoV CP with the 2A linker peptide (PMMoV-GFP^2A^) and used it to visualize and follow the spread of the virus into new leaves. Electron microscopy and immunocolloidal gold confirmed that PMMoV-GFP^2A^ could form rod-shaped particles displaying GFP. Using this tool, we were able to demonstrate that the CP is located in the cytoplasm 4 days after systemic infection and then accumulates in irregular patches near chloroplasts at 8 days. The self-assembly of virions was confirmed using a combination of confocal laser scanning and electron microscopy. Thus, PMMoV-GFP^2A^ is a useful tool for studying the spatial and temporal changes in PMMoV CP during viral infection.

## 2. Results

### 2.1. The Constructed PMMoV-GFP^2A^ Vector Is Capable of Infecting Plants under Normal Conditions

PMMoV-GFP^2A^ was constructed based on the infectious PMMoV-GFP virus clone previously created which expresses free GFP [10]. As with the related TMV (30B) [18], the free GFP of PMMoV-GFP is translated from the CP subgenomic mRNA promoter. In PMMoV-GFP^2A^, the 2A linker was fused between GFP and CP, and this vector can form free CP, free GFP, and a GFP-2A-CP fusion protein. To compare the infection efficiency of the PMMoV, PMMoV-GFP, and PMMoV-GFP^2A^ clones (Figure 1A), the vectors were electroporated into *Agrobacterium tumefaciens*, which were subsequently delivered to *Nicotiana benthamiana*. At 2 days post systemic infection (dpi), a UV lamp examination revealed GFP fluorescence in the systemic leaves of PMMoV-GFP and PMMoV-GFP^2A^. By 4 dpi, differences in fluorescence distribution were observed in the systemically infected leaves of PMMoV-GFP (mosaic) and PMMoV-GFP^2A^ (plaque) (Appendix A). In all three treatments, severe crinkling of systemic leaves was observed at 8 dpi. The UV lamp examination showed GFP fluorescence in the upper non-inoculated leaves of PMMoV-GFP and PMMoV-GFP^2A^, but not in those infected by PMMoV (Figure 1A). Western blots showed that the coat protein could be detected in the new non-inoculated leaves of all infected plants (Figure 1C). After inoculation with PMMoV-GFP^2A^, the Western blot detected bands at the anticipated molecular weights of 17.2 and 49.1 kDa. The GFP-specific antibody also revealed two bands corresponding to the free GFP and the fusion protein (GFP-2A-CP) (Figure 1C). Absolute quantification PCR assays showed that at 2, 4, 6, and 8 dpi, the virus titers were greatest following inoculation with PMMoV. At 2 and 4 dpi, virus titers were greater following inoculation with PMMoV-GFP than with PMMoV-GFP^2A^, but at 6 and 8 dpi, there were no significant differences in titers between PMMoV-GFP and PMMoV-GFP^2A^ inoculations (Figure 1D).

### 2.2. The Virus Particles Produced by PMMoV-GFP^2A^ Incorporate GFP

To further demonstrate the integrity of our constructed vectors, the virions produced by them in systemically infected leaves of *N. benthamiana* were observed at 8 dpi. Virus particles were extracted, purified, negatively stained, and then observed using TEM. Typical rod-like particles were observed following inoculation by each of the vectors (Figure 2A), and these all had lengths mostly in the 280~320 nm range and diameters of 17~18 nm. This similarity in morphology may explain the ability of modified virus particles to infect the systemic leaves normally and produce symptoms. To determine if the GFP-2A-CPs were assembled into virus particles and displayed on the surface, the extracted virus particles were labelled with antibodies specific for PMMoV CP or GFP using immunogold labelling. CP-specific labeling was observed around all three types of virus particle but GFP-specific labeling occurred only around PMMoV-GFP^2A^ (Figure 2B). The number of colloidal gold particles surrounding and within the field of view of the viral particles was quantified and analyzed to determine the ratio of labelling for both antibodies around the viral particles. The CP-specific antibody-labelled viral particles produced by PMMoV and PMMoV-GFP had a labelling ratio exceeding 90%, with less than 0.2% labelled with the GFP antibody, implying that both viral particles use available CP for assembly. In contrast, CP labelled PMMoV-GFP^2A^ at a slightly lower proportion than the other two, approximately 70%, and the labelling rate of GFP reached approximately 50% (Figure 2C). This provides strong evidence that PMMoV-GFP^2A^ virus particles contain both the free CP and the GFP-2A-CP protein.

### 2.3. PMMoV-GFP^2A^ Can Track the Dynamic Changes of Virus Coat Protein by Tracking GFP

We next delivered the two virus vectors expressing GFP to *N. benthamiana* plants using infiltration and compared the fluorescence in systemic leaves using confocal microscopy 4 and 8 days after systemic infection, using the transient expression of GFP-2A-CP as a control (inoculated leaves at 4 days). With both viruses (PMMoV-GFP and PMMoV-GFP^2A^), fluorescence was predominantly cytoplasmic at 4 days and similar to that of the transient GFP-2A-CP control (Figure 3A). At 8 days, the fluorescence distribution pattern remained similar for PMMoV-GFP, but with PMMoV-GF^P2A^ there was a noticeable aggregation of fluorescence at chloroplasts and within the cytoplasm (Figure 3B). As PMMoV-GFP^2A^ virions had incorporated GFP proteins on their surface (Figure 2), the emergence of these fluorescent spots appears likely to be caused by the aggregation of virions.

To validate this observation, ultrathin sections of leaves harvested 8 days after systemic infection by the three viruses (PMMoV, PMMoV-GFP, and PMMoV-GFP^2A^) were observed by means of electron microscopy, using leaves inoculated with empty vector (pCB301) as a control. The chloroplast morphology of all four samples was similar but virion aggregations in vesicles were frequently observed around the chloroplasts in all three virus samples but not in the controls (Figure 4A,B). To ascertain whether the virion aggregations could correspond with fluorescence plaques, three virus-infected leaves were labelled with antibodies specific for PMMoV CP or GFP using immunogold labelling. Similar to the immunogold labeling of virions, CP-specific labeling was observed around all three types of virus particle, but GFP-specific labeling occurred only around PMMoV-GFP^2A^ (Figure 4C,D). This strongly suggests that the large fluorescent plaques were the result of virion aggregation.

### 2.4. Virions or Virion Aggregation Can Form Fluorescent Plaques

Carboxylate groups have been found at subunit interfaces in the structures of many plant viruses, often forming calcium binding sites [9]. In the coat protein of TMV, the amino acids Glu50 and Asp77 have been proposed as carboxylate groups. When the lateral interacting carboxylate groups were replaced with neutral residues (E to Q, D to N), rod-shaped particles self-assembled independently of the genomic RNA of TMV [25]. The alignment of the CP amino acid (aa) sequences of PMMoV and TMV revealed that aas 52~ 54, 60~64, 68~70, and 72~77 of PMMoV are conserved with those of TMV (Figure 5A). The D51 and D78 of PMMoV-CP are negatively charged, similar to the E50 and D77 of TMV-CP. To confirm that these two negatively charged amino acids control virion assembly, a PMMoV CP expression vector was constructed with the mutations D51Q and D78N. This vector (pET28a-CP_D51Q,D78N_) and the control pET28a-CP (Figure 5B) were then transformed into chemically competent BL21-CodonPlus cells via heat shock. After the culturing, protein expression, and purification of virus-like particles (VLPs) as described [29], only pET28a-CP_D51Q,D78N_ formed a band in cesium chloride density-gradient centrifugation (Figure 5C). The band was removed with a syringe, precipitated using centrifugation, and resuspended. Short rod-shaped particles were then observed using transmission electron microscopy (Figure 5C).

A previous study with TMV showed that peptides and proteins can be displayed from the C terminus of the TMV CP with the inclusion of a (GGGGS)n (*n* ≤ 3) linker peptide [23,24]. We therefore created a transient eGFP expression vector (CP_D51Q,D78N-eGFP_), in which eGFP was fused to the C terminal of PMMoV CP_D51Q,D78N_ via a (GGGGS)_3_ linker peptide. This transient expression vector, and CP-eGFP as control, were electroporated into *A. tumefaciens* and then delivered to *N. benthamiana* plantlets using infiltration. At 4 dpi, green fluorescent granules were observed in leaves inoculated with CP_D51Q,D78N_-eGFP, but not in the controls (Figure 5D). After PEG precipitation of inoculated leaf extracts, rod-like particles were observed in those inoculated with CP_D51Q,D78N_-eGFP but not in the controls (Figure 5E). This suggests that GFP-tagged virions can be formed and that aggregations could appear as fluorescent plaques in confocal laser microscopy.

## 3. Discussion

Tobamoviruses are positive single-stranded RNA viruses capable of encoding at least four proteins: the replicase proteins (126 kDa and 183 kDa), directly translated from the genomic RNA; and the movement protein (MP, 30 kDa) and coat protein (CP, 17 kDa), translated from subgenomic (sg) promoters [30]. The full and minimal MP sgRNA promoters are localized between −95 and +40 and −35 and +10, respectively (relative to the transcription start site). The minimal and full CP sgRNA promoters are identified in positions −69 to +12 and −157 to +54, respectively [31]. Based on these studies, a strategy for expressing foreign genes involves replacing the CP ORF with a reporter gene ORF, then adding a second tobamovirus CP subgenomic mRNA promoter followed by a CP ORF [32]. Within the tobamovirus vectors, the most efficient vector (TMV:30B) has been identified, which replaces the CP with a foreign protein and adds the CP subgenomic mRNA promoter, CP ORF, and 3′ nontranslated region from tobacco mild green mosaic virus (TMGMV) U5 [18].

In this study, the infectious PMMoV-GFP virus clone has a cassette inserted (GFP, 3′ UTR of TMV and the CP subgenomic mRNA promoter of TMGMV) between MP and CP. The PMMoV-GFP^2A^ clone has a cassette inserted (GFP and 2A) between MP and CP. At 2 and 4 dpi, there are significant differences in virus titers in the following order: PMMoV > PMMoV-GFP > PMMoV-GFP^2A^ (Figure 1D). The incomplete matching and partial deletion of the CP sgRNA promoter (PMMoV-GFP and PMMoV-GFP^2A^, respectively) could decrease the infections. Additionally, Western blot and RT-qPCR analyses reveal that the accumulation levels of the CP proteins and viral mRNA of PMMoV-GFP are much higher than those of PMMoV-GFP^2A^ at the same time (Figure 1C,D), but the fluorescence of PMMoV-GFP^2A^ is much brighter than PMMoV under UV light. The visual differences could be caused by the differences in GFP expression strategies of PMMoV-GFP and PMMoV-GFP^2A^. At 4 dpi, the fluorescence distributions of PMMoV-GFP and PMMoV-GFP^2A^ are mosaic and plaques, respectively (Appendix A). The GFPs of PMMoV-GFP^2A^ are displayed on the surface of virions and form fluorescent plaques in cells, leading to the observation of fluorescent plaques in systemic infection leaves. However, this is not observed in PMMoV-GFP systemic infection leaves (Figure 1B).

Virus-based vectors are beginning to have a major impact as expression systems for the production of specialty products. Tobamoviruses, for instance, can constitute as much as 10% of the dry weight of a plant [33]. Consequently, a series of tobamovirus-based hybrid vectors for the expression of foreign genes is having a major impact on research, biotechnology, and therapeutic treatments [18,19,20,24]. More than 2100 copies of tobamovirus CP can form rigid, rod-shaped viral particles (300 nm × 18 nm) and can be purified industrially by using simple protocols [34]. The foreign protein is fused with the N- or C-terminal of tobamovirus CP through genetic reprogramming. The foreign protein is fused either N- or C-terminally with a linker peptide (flexible glycine-rich linker (GGGGS)_3_ or helical linker (EAAAK)_3_). Both N- and C-terminal fusion proteins containing either linker are expressed at a high level, but only displayed on the surface of virion as a C-terminal fusion to CP via a short flexible linker [24]. When the foreign protein is fused via 2A with the N-terminal of CP, it can form the virion and be displayed on the surface [23]. This approach allows for the simultaneous production of foreign proteins fused to the CP, free foreign protein, and free CP subunits. This suggests that free CP subunits might be crucial for viral particle assembly when the foreign protein is fused with the N-terminal of CP. In this study, PMMoV virions can be directly imaged as fluorescent plaques using confocal laser microscopy when GFP is fused via 2A with the N-terminal of CP. The foreign protein can form viral particles and be displayed on the surface as a C-terminal fusion to CP via a short flexible linker (Figure 5), but prevents the systemic movement of the virus [23]. Although PMMoV can be transported over long distances even without the presence of the CP protein [14], the free CP subunits could be a curial factor for the viral systemic infection efficiency.

The CP of tobamoviruses is a multifunctional protein, which participates in encapsidation, viral translation, replication, cell to cell movement, and systemic movement [3,8,22]. The immunofluorescent labeling of TMV-infected protoplasts shows that CP enhances the production of MP and increases the size of the VRC, but the dynamic changes of CP are unknown [17]. The heterologous expression of GFP in a chimeric virus is a common tool for studying the systemic movement of plant viruses. The expression of free GFP has been widely used and is useful for identifying infected tissues, but whether virions can be directly labelled when GFP is expressed as a fusion with the viral CP is unknown [14,18,24,27,35]. Some genetically modified vectors have been developed that display GFP on the surface of the virion as N- or C-terminal coat protein fusions through the integration of the 2A sequence, including TMV, potato virus X (PVX), broad bean wilt virus 2, cowpea mosaic virus [36,37,38], and PMMoV in this study. Using this strategy, GFP-tagged virion aggregations of PMMoV and PVX can be imaged as fluorescent plaques and fibrillar aggregates using confocal laser microscopy (Figure 3 and Figure 5) [35]. It is not clear whether fluorescent plaques can be observed with the other viruses.

In the field of human virology, the real-time tracking of single virus particles in vivo has emerged as a prominent area of research. The diminutive size of virions poses a challenge for direct fluorescence tracing through confocal laser microscopy. Nevertheless, quantum dots have proven effective when encapsulated in Human Immunodeficiency Virus type 1 particles, enabling the tracking of viral entry at a single-particle level in live human primary macrophages [39]. Regrettably, the plant cell wall acts as a natural protective barrier, impeding the direct application of most nanoparticles. In the present study, PMMoV-GFP^2A^ demonstrates the capability to form rod-shaped particles that prominently display GFP. The resulting aggregations of GFP-tagged virions can be visualized as fluorescent plaques through confocal laser microscopy. Consequently, PMMoV-GFP^2A^ serves as a valuable tool for investigating the spatial and temporal dynamics of PMMoV CP during viral infection and may prove instrumental in elucidating the mechanisms underlying virion assembly in vivo.

## 4. Materials and Methods

### 4.1. Plasmid Construction

The modified PMMoV-GFP^2A^ was based on the infectious virus clone previously created [10]. The vector of PMMoV contained, in sequential order, a 2× 35S; a full length sequence of PMMoV (GenBank: MN616926.1); a nucleotide sequence of a hepatitis delta virus (HDV) ribozyme and NOS terminator. Based on PMMoV, a group of elements was inserted into sequences of PMMoV MP and CP. The element box contained an mGFP gene sequence (GenBank: U87974.1); the 3′-UTR of TMV (nt 9~182, GenBank: MN186255.1); and the subgenomic promoter of the tobacco mild green mosaic virus CP (nt 5496~5658, GenBank: MK005155.1). PMMoV-GFP^2A^ was fused with the mGFP gene sequence (without the stop codon) and a linker peptide sequence (2A: 5′-CAGCTGTTGAATTTTGACCTTCTTAAGCTTGCGGGAGACGTCGAGTCCAACCCTGGGCCC-3′) at the N-terminus of PMMoV CP.

PCR amplification was performed using KOD-PlusNeo-FX (TOYOBO, Osaka, Japan), and homologous recombinants of all constructs were made using the CloneExpress MultiS One Step Cloning Kit (Vazyme, Nanjing, China), with the detailed methods described before [40]. The primers used in this study are listed in Table 1.

### 4.2. Plant Infection and Fluorescence Detection

*Nicotiana benthamiana* plants were cultured in a light incubator with 14 h light and 8 h dark at 26 °C. Plants at the six-leaf stage were inoculated with three constructs, namely, PMMoV, PMMoV-GFP, and PMMoV-GFP^2A^, and blank vector pCB301 (as control). Photos were taking under bright field natural light and under UV light.

### 4.3. Western Blot

Total proteins of *N. benthamiana* leaves (100 mg) were extracted with protein extraction buffer (50 mmol L^−1^ sodium phosphate buffer pH 7.0, 5 mmol L^−1^ β-mercaptoethanol, 10 mmol L^−1^ EDTA, 0.1% Triton X-100) (Sigma-Aldrich, St. Louis, MO, USA), mixed with 5× loading buffer, and separated using 12% SDS polyacrylamide gel electrophoresis (PAGE). The proteins were transferred onto nitrocellulose membranes (Amersham, London, OH, USA) using transfer buffer (48 mmol L^−1^ of Tris base, 39 mmol L^−1^ of glycine, and 20% methyl alcohol). Rabbit anti-CP and mouse anti-GFP primary antibodies (Sigma-Aldrich, St. Louis, MO, USA), diluted 1:5000, followed by horseradish peroxidase-coupled goat anti-Rabbit/mouse IgG (Sigma-Aldrich, St. Louis, MO, USA), diluted 1:5000, were used for Western blot.

### 4.4. Quantitative Real-Time PCR

Total RNA was extracted from systemic leaves of plants infected by PMMoV, PMMoV-GFP, or PMMoV-GFP2A using Trizol reagent (Invitrogen, Waltham, MA, USA). The total RNA was treated with DNase I to remove genomic DNA. cDNA synthesis was conducted using the ReverTra Ace Kit (TOYOBO, Osaka, Japan) and random primers. qPCR was performed by using the TOROGreen^®^ qPCR Master Mix Kit (TOYOBO, Osaka, Japan). The cycling conditions for the subsequent PCR were as follows: 95 °C 5 min, and then 40 cycles of 95 °C for 15 s, 58 °C for 20 s, and 72 °C for 20 s. Equal-proportion dilutions (10^−2^, 10^−3^, 10^−4^, 10^−5^, and 10^−6^) of PMMoV plasmid were used to create a standard curve from which sample concentration was determined. The primers (RdRp f and RdRp r) used for qPCR are listed in Table 1.

### 4.5. Virus Preparation and Electron Microscopy

A total of 100g of systemic leaves of virus-infected *N. benthamiana* was blended and stirred in 0.2M phosphate buffer (pH 7.2) in the ratio of 1:3; then, 8% chloroform was added and the mixture was stirred for 30min at room temperature until chlorophyll deposition. After centrifuging at 8000 rpm, 4 °C for 20min, the supernatant was aspirated and mixed with PEG8000 (6%) and NaCl (3%), stirred for 1 h at 4 °C, and then precipitated at 4 °C for more than 4 h. The supernatant was then centrifuged at 4 °C, 8000 rpm for 20 min, and the precipitate resuspended overnight in 0.01 M phosphate buffer (pH 7.2) at 4 °C. After centrifugation at 5000 rpm for 10 min at 4 °C, the supernatant was collected and extracted with 8% chloroform to remove PEG. Then, 3 mL of 25% sucrose pad was added to the bottom of the ultracentrifuge tube and centrifuged at 40,000 rpm for 4 h. The resulting precipitate was suspended in ultrapure water, providing the crudely purified virus particles. The viral extract was diluted 100 times, the OD value at A260 was measured, and the concentration of purified virus was calculated according to the following formula: virus concentration (mg/mL) = OD_260_ × dilution times ÷ extinction coefficient (3.1). The purified virions were absorbed on hydrophilic copper grids and negatively stained with a 2% phosphotungstic acid for TEM observation.

### 4.6. Ultrathin Sectioning

Small pieces (1 mm × 1 mm) of infected systemic leaves were quickly immersed in 2.5% glutaraldehyde solution for fixation at room temperature for 2 h. After washing for three times with phosphate buffer (PB), fixing with 1% OsO_4_ for 1 h, and rinsing three times with PB, the samples were dehydrated using a 50%, 70%, 80%, 90%, and 95% ethanol gradient, followed by 100% ethanol for 20 min and 100% acetone for 20 min. The Spurr embedding agent was mixed with acetone 1:1 and then permeabilized for 1 h. The embedding agent was mixed with acetone 3:1 and permeabilized for 3 h. After polymerization at 70 °C for 24 h, the embedded samples were sectioned using a Leica EM UC7 (Leica, Wetzlar, German) ultrathin sectioning machine to obtain 70 nm sections. The sections were stained with uranium acetate and lead citrate, and then observed under a Hitachi-7800 transmission electron microscope (Hitachi, Tokyo, Japan).

### 4.7. Immunocolloidal Gold Technique

The nickel mesh covered with virions was incubated with specific antibodies for 5 min and rinsed three times with distilled water. After blocking with 1% BSA for 30 min and rinsing three times with distilled water, the samples were incubated with colloidal gold-labeled antibody for 45min. Unlabeled antibody was removed by washing with distilled water, and the samples were then stained with phosphotungstic acid and observed under the HT-7800 TEM.

### 4.8. Immuno Electron Microscopy

In the electron microscopy process, samples of leaves infected with the virus were subjected to fixation, dehydration, and embedding in LR GOLD resin. For immunogold labeling, ultrathin sections underwent incubation with either CP or GFP Antibody. Subsequently, immunogold-labeled goat antibodies against rabbit IgG, conjugated with 15 nm gold particles (ABcam, London, UK), were applied. The sections were then visualized using a transmission electron microscope HT7800 (Hitachi, Tokyo, Japan).

### 4.9. Virus-like Particle Purification

Virus-like particles (VLPs) were prepared using a modified method as previously reported [29]. The plasmids pET28a-CP or CP_D51Q,D78N_ were introduced into BL21-CodonPlus cells. A single colony was inoculated into 5 mL Luria–Bertani (LB) medium and incubated at 37 °C for 3 h. Subsequently, 200 µL of bacterial culture was inoculated into 50 mL prewarmed medium and placed in a 37 °C incubator, shaking at 150 rpm. The culture was grown to an OD600 = 0.5, followed by the addition of 50 µL of 1 M IPTG. The incubation continued overnight at room temperature with shaking at 150 rpm. After the overnight incubation, the culture was transferred to a 50 mL polypropylene conical tube and centrifuged at 16,000× *g* for 10 min, and the supernatant was poured off. Subsequently, 10 mL of 0.1 M phosphate buffer was added, and the mixture was incubated at room temperature with gentle shaking (20 rpm) until the pellet fully dissolved. Next, 10 µL of Lysozyme solution (50 mg/mL; Solarbio, Beijing, China) was added, and the solution was incubated for 5 min at room temperature with gentle shaking. The dissolved pellet was transferred to screw-cap polycarbonate (25 × 89 mm) ultracentrifuge tubes and centrifuged at 91,000× *g* for 1 h at 4 °C. The supernatant was then poured off, and 4 mL of 0.1 M phosphate buffer was added to the screw-cap ultracentrifuge tubes to dissolve the VLP-containing pellet overnight. The following morning, 2 mL of the resuspended pellet was gently layered onto the top of the 10–40% Cesium chloride (CsCl) density gradient. The gradient was centrifuged at 91,000× *g* for 1.5 h at 4 °C. Using a syringe with a cannula, the VLP band was carefully removed and transferred to a screw-cap ultracentrifuge tube. The tube was then centrifuged at 92,000× *g* at 4 °C for 1.5 h to pellet the VLPs. The supernatant was carefully discarded, and the pellet was gently resuspended in 0.2–1 mL of 0.1 M phosphate buffer at pH 7.0.

## Figures and Tables

**Figure 1 plants-13-00928-f001:**
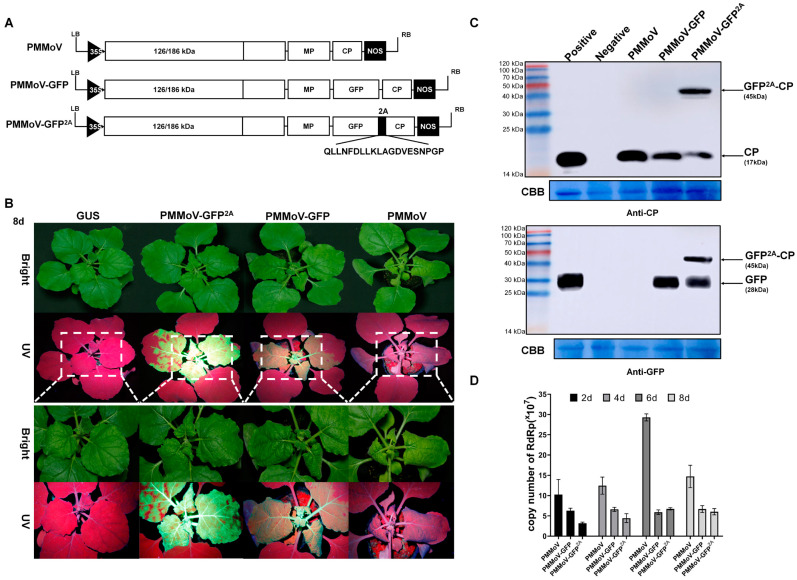
Construction of chimera virus vectors. (**A**) Schematic representation of chimera constructions (PMMoV, PMMoV-GFP, and PMMoV-GFP^2A^). (**B**) Symptoms in *N. benthamiana* plants inoculated with PMMoV, PMMoV-GFP, and PMMoV-GFP^2A^ under UV and natural light at 8 dpi. (**C**) Western blot analysis confirming the presence of CP and GFP proteins in systemic leaves of *N. benthamiana* plants. Negative, healthy plant; Positive, leaves transiently expressing CP or GFP; CBB, Coomassie Brilliant Blue. (**D**) Absolute quantification PCR assays showing the copy numbers of PMMoV, PMMoV-GFP, and PMMoV-GFP^2A^ at 2, 4, 6, and 8 dpi.

**Figure 2 plants-13-00928-f002:**
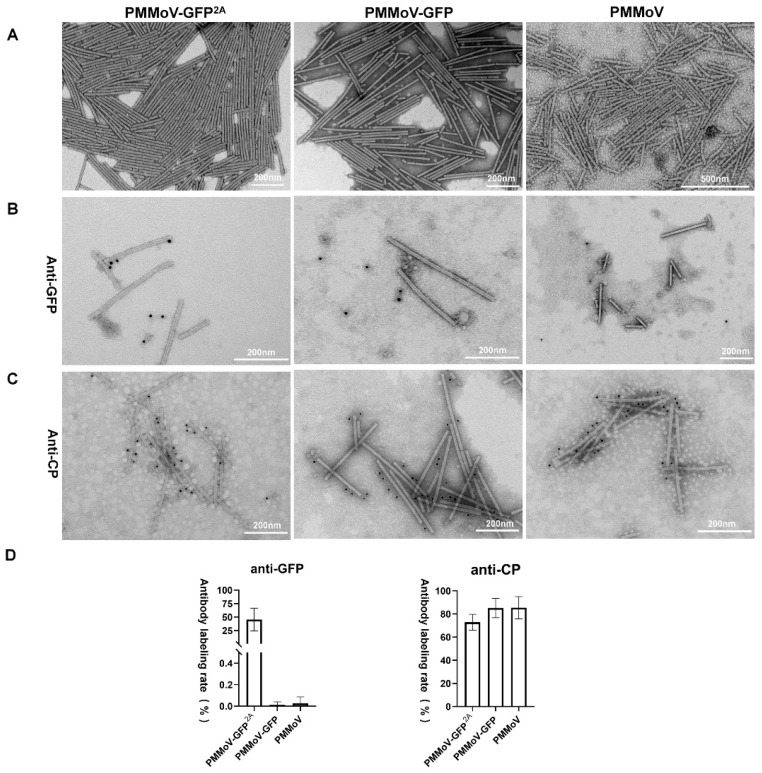
Properties of virus particles of PMMoV-GFP^2A^, PMMoV-GFP, and PMMoV. (**A**) Morphological observation of virus particles in the electron microscope. (**B**,**C**) Immune colloidal gold labelling of virions with GFP antibody or CP antibody. (**D**) Rates of CP- or GFP-antibody labeling.

**Figure 3 plants-13-00928-f003:**
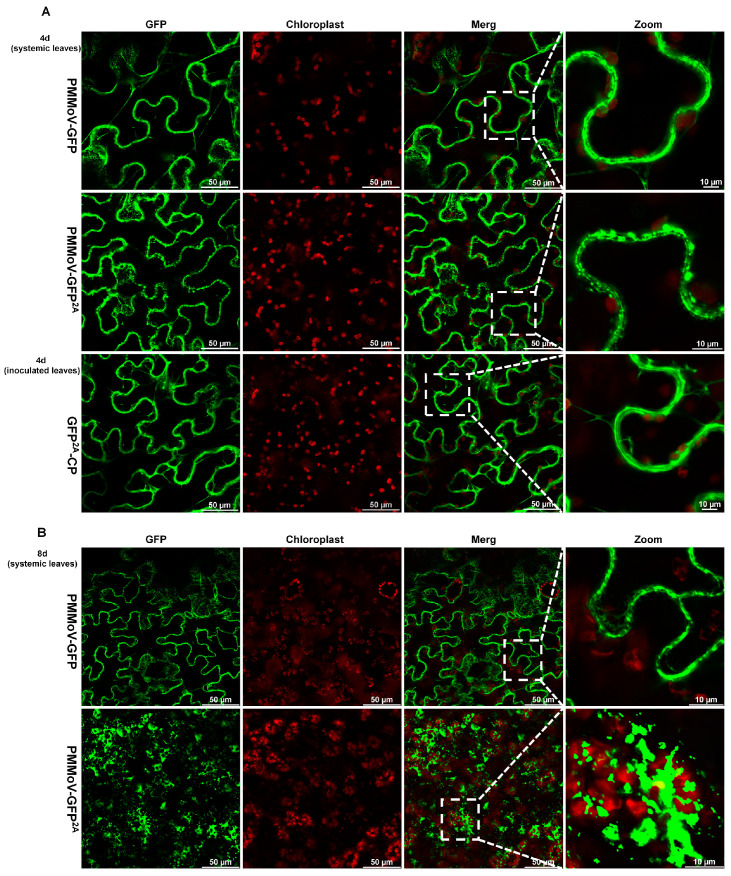
Confocal laser microscopy images of systemic leaves of plants after inoculation with PMMoV-GFP and PMMoV-GFP^2A^ or transiently expressing GFP^2A^-CP. (**A**) Four days after systemic infection. (**B**) Eight days after systemic infection.

**Figure 4 plants-13-00928-f004:**
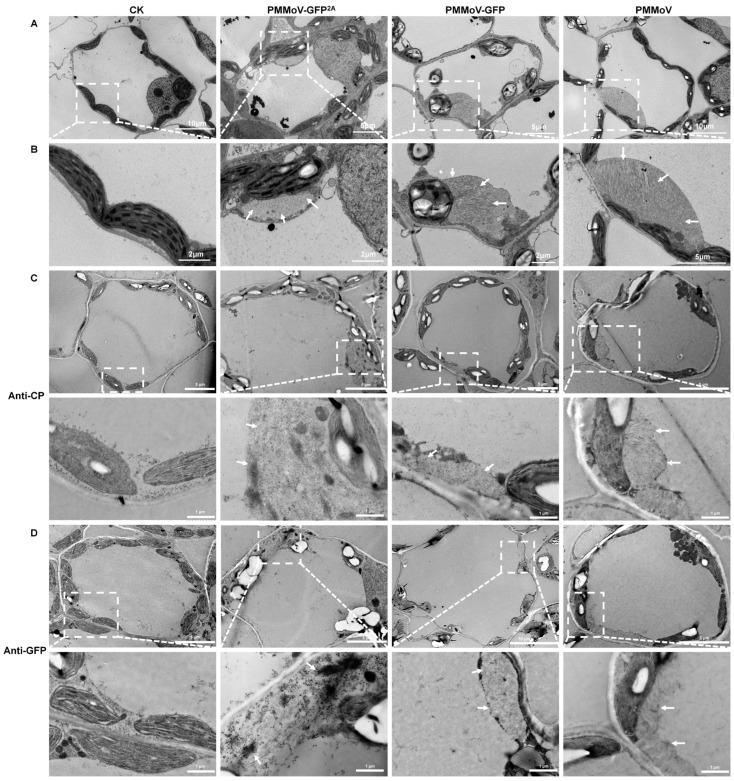
Transmission electron micrographs showing the ultrastructure of virus-infected leaves. (**A**) General images 8 days after systemic infection. Healthy plants as control (CK). (**B**) Enlargements showing virion aggregations (marked by white arrows). (**C**,**D**) Immune colloidal gold labelling of virions with CP antibody and GFP antibody. Enlargements showing virion aggregations (marked by white arrows; enlarged regions were marked by white square).

**Figure 5 plants-13-00928-f005:**
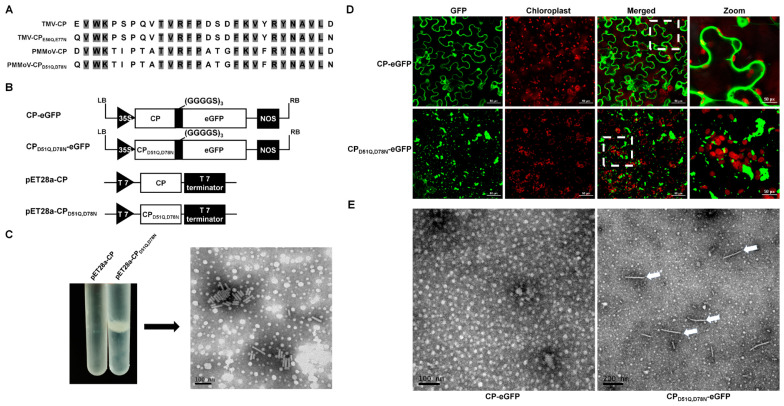
Incorporation of GFP into virions. (**A**) Partial alignment of CP amino acid sequences of PMMoV and TMV showing the conserved self-assembly sites. (**B**) Schematic representation of self-assembly vectors and fluorescent transient expression vectors. (**C**) Cesium chloride (CsCl) density-gradient centrifugation and transmission electron micrograph of purified PMMoV VLPs. (**D**) Fluorescence of CP-eGFP and CP_D51Q_,_D78N-eGFP_ 4 days after inoculation. (**E**) Transmission electron micrograph of purified extracts of leaves inoculated with CP-eGFP or CP_D51Q,D78N_-eGFP showing VLPs (arrows).

**Table 1 plants-13-00928-t001:** Primers used in this study.

Target	Name	Sequence (5′-3′)
PMMoV-GFP	Vec-PMMoV-GFP f	ATGGCTTACACAGTTTCCAGTG
	Vec-PMMoV-GFP r	GATTAAGCACAGTATAAACG
	PMMoV-GFP-1 f	CGTTTATACTGTGCTTAATC
	PMMoV-GFP-1 r	TTAAAACGAAGAAGACTCGGCG
	PMMoV-GFP-2 f	CGCCGAGTCTTCTTCGTTTTAACTATGAAGACTAATCTTTTTC
	PMMoV-GFP-2 r	CACTGGAAACTGTGTAAGCCATAGTTAAAACGTACTCGATGAC
PMMoV-GFP^2A^	PMMoV-GFP^2A^-2 f	GCCGAGTCTTCTTCGTTTTAACTATGAAGACTAATCTTTTTCTC
	PMMoV-GFP^2A^-2 r	CACTGGAAACTGTGTAAGCCATGGGCCCAGGGTTGGACTCGAC
GFP^2A^-CP	GFP^2A^-CP f	TTGGAGAGAACACGGGGGACGAGCTCATGAAGACTAATCTTTTTCTC
	GFP^2A^-CP r	AACGAAAGCTCTGCAGTCTAGATTAAGGAGTTGTAGCCCAGGTGAG
pET28a-CP/ pET28a-CP_D51Q,D78N_	28a-CP f	TTGTTTAACTTTAAGAAGGAGATATACCATGGCTTACACAGTTTCCAGTG
	28a-CP r	CAGTGGTGGTGGTGGTGGTGCTCGAGCTAAGGAGTTGTAGCCCAGGTG
CP-eGFP/ CP_D51Q,D78N_-eGFP	CP-eGFP-1 f	TTGGAGAGAACACGGGGGACGAGCTCATGGCTTACACAGTTTCC
	CP-eGFP-1 r	CCCGAGCCACCGCCACCGGTACCAGGAGTTGTAGCCCAGGTGAGTCC
	CP-eGFP-2 f	ACCGGTGGCGGTGGCTCGGGCGGTGGTGGGTCGGGTGGCGGTGGCTCG GGATCCATGGTGAGCAAGGGCGAGGA
	CP-eGFP-2 r	GAACGAAAGCTCTGCAGTCTAGATTACTTGTACAGCTCGTCCAT
Real-time qPCR	RdRp f	AAGAAAGGAGGTTATGGTCAGC
	RdRp r	CGGCAGTAATCTCATCCCAG

f: forward primer; r: reverse primer.

## Data Availability

The data presented in this article are available on request from the corresponding authors.

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
