# Peer review of "Adoption of the 2A Ribosomal Skip Principle to Track Assembled Virions of Pepper Mild Mottle Virus in Nicotiana benthamiana"

_plants, 2024, doi:10.3390/plants13070928_

Round 1

Reviewer 1 Report

Comments and Suggestions for Authors

Dear Authors,

Firstly, I have concerns about the paper, as it is not clearly demonstrated and lacks sufficient experimental data. In its current form, the paper is not acceptable for publication. I urge the authors to carefully consider my comments and address all of them. Please see the comments below.

Given that similar work has already been published on pepper mottle mosaic virus, the authors should clearly highlight the novelty of their paper. 10.1007/s11033-018-4449-4. (Development of a new tobamovirus-based viral vector for protein expression in plants)

Authors must conduct the following experiments: Immuno capture electron microscope Assay for confirming GFP and VLPs expression in a mixture of wild and recombinant purified particles. This will help readers clearly understand the expression of GFP and VLPs.

The article was not well-written, and there are inconsistencies in citation formatting, with references sometimes cited by numbering and other times by "et al." The authors should carefully review the entire paper before submitting.

The western blotting images are not satisfactory, and the authors should provide clearer ones.

The western blot confirmation of GFP fused with CP using PMMoV-GFP does not show the fused protein, but plants were showing GFP expression. This discrepancy needs to be addressed.

Author Response

Authors’ response to comments received

We thank the editor and the reviewers for their constructive comments and have now modified the manuscript to correct the errors and to respond to the comments and suggestions. In the text below, the comments received are in black and our replies and revisions in red.

Reviewer1:

Firstly, I have concerns about the paper, as it is not clearly demonstrated and lacks sufficient experimental data. In its current form, the paper is not acceptable for publication. I urge the authors to carefully consider my comments and address all of them. Please see the comments below.

Given that similar work has already been published on pepper mottle mosaic virus, the authors should clearly highlight the novelty of their paper. 10.1007/s11033-018-4449-4. (Development of a new tobamovirus-based viral vector for protein expression in plants)

Reply: We appreciate the valuable feedback from the reviewer. The PMMoV-based vector system for high-level production of heterologous proteins in plants has indeed been previously documented (doi:10.1007/s11033-018-4449-4). However, the specific procedures for constructing a chimeric virus aimed at investigating the systemic movement or labeling of PMMoV virions have not been adequately discussed in the existing literature. Our current study addresses this gap by detailing the construction of a series of chimeric viral vectors, facilitating the selection of an efficient vector for investigating the spatial and temporal dynamics of PMMoV CP during viral infection.

Authors must conduct the following experiments: Immuno capture electron microscope Assay for confirming GFP and VLPs expression in a mixture of wild and recombinant purified particles. This will help readers clearly understand the expression of GFP and VLPs.

Reply: We have added this experiment in Figure 4C-D.

The article was not well-written, and there are inconsistencies in citation formatting, with references sometimes cited by numbering and other times by "et al." The authors should carefully review the entire paper before submitting.

Reply: Corrections have been made in Lines 224 and 229.

The western blotting images are not satisfactory, and the authors should provide clearer ones.

Reply: The experiment has been repeated and the results are reflected in the revised Figure 1C.

The western blot confirmation of GFP fused with CP using PMMoV-GFP does not show the fused protein, but plants were showing GFP expression. This discrepancy needs to be addressed.

Reply: In this study, we have construct two chimeric viral vectors: PMMoV-GFP and PMMoV-GFP2A. PMMoV-GFP is capable of expressing free CP and free GFP, whereas PMMoV- GFP2A can produce free CP, free GFP and GFP-2A-CP fusion protein. The GFP-specific antibody of PMMoV-GFP and PMMoV-GFP2A revealed a single band (indicating free GFP) and two bands (representing free GFP and GFP-2A-CP), respectively (Figure 1C).

Reviewer 2 Report

Comments and Suggestions for Authors

The ms. by Mengting et al. is a technical piece providing support to the use of 2A peptide in plant virology. The ms. also backs useful data on viral systemic movement, tobamovirus-plant interaction, sub-cellular distribution of tobamovirus CP, and structural-function relationship in that protein. Although several of these aspects are already known from studies of other tobamoviruses, the ms. deserves to be published. The study is well-described and brings a detailed description of the methodology. The figures are well organized and support the obtained results. However, the discussion section is weak, it seems an extension of the introductory section. In my opinion, it needs to be reformulated because its current format drastically reduces the quality of the work, and hinders its acceptance. The Conclusion section does not add clarity. Consider removing.

Some other minor issues to be amended:

Explain “VRC” in the introduction. Virus replication complex.

Explain “CBB” in the legend of Figure 1, and “CK” in Figure 4.

Section 2.4. Scientific names in italics, i.e., A. tumefaciens, N. benthamiana.

Add the city of the headquarters for all the suppliers (Amersham or GE biosciences) and instruments (Hitachi, Leica, etc.) listed in the M&M section.

Table 1. In the heading replace Objective with Target, name with Name.

For the sake of clarity, at the end of section 4.4, indicate the pair of primers used for virus load quantification.

Line 114. “Surr” or Spurr embedding agent.

Figure 1. GFP fluorescence from PMMoV- GFP2A seems to be markedly more intense than from the PMMoV-GFP. However, data shown in Figures 1D and E seem not to support a greater accumulation of PMMoV- GFP2A. Any idea about this?

Author Response

Authors’ response to comments received

We thank the editor and the reviewers for their constructive comments and have now modified the manuscript to correct the errors and to respond to the comments and suggestions. In the text below, the comments received are in black and our replies and revisions in red.

Reviewer 2:

The ms. by Mengting et al. is a technical piece providing support to the use of 2A peptide in plant virology. The ms. also backs useful data on viral systemic movement, tobamovirus-plant interaction, sub-cellular distribution of tobamovirus CP, and structural-function relationship in that protein. Although several of these aspects are already known from studies of other tobamoviruses, the ms. deserves to be published. The study is well-described and brings a detailed description of the methodology. The figures are well organized and support the obtained results. However, the discussion section is weak, it seems an extension of the introductory section. In my opinion, it needs to be reformulated because its current format drastically reduces the quality of the work, and hinders its acceptance. The Conclusion section does not add clarity. Consider removing.

Reply: We appreciate the valuable feedback from the reviewer. Following your suggestions, we have expanded the discussion section and omitted the conclusion.

Some other minor issues to be amended:

Explain “VRC” in the introduction. Virus replication complex.

Reply: Added.

Explain “CBB” in the legend of Figure 1, and “CK” in Figure 4.

Reply: Added in Lines 146-147 and 217-218.

Section 2.4. Scientific names in italics, i.e., A. tumefaciens, N. benthamiana.

Reply: Corrected.

Add the city of the headquarters for all the suppliers (Amersham or GE biosciences) and instruments (Hitachi, Leica, etc.) listed in the M&M section.

Reply: Corrected.

Table 1. In the heading replace Objective with Target, name with Name.

Reply: Corrected.

For the sake of clarity, at the end of section 4.4, indicate the pair of primers used for virus load quantification.

Reply: Corrected.

Line 114. “Surr” or Spurr embedding agent.

Reply: Corrected.

Figure 1. GFP fluorescence from PMMoV- GFP2A seems to be markedly more intense than from the PMMoV-GFP. However, data shown in Figures 1D and E seem not to support a greater accumulation of PMMoV- GFP2A. Any idea about this?

Reply: At 4 dpi, the fluorescence distributions of PMMoV-GFP and PMMoV-GFP2A exhibit mosaic and plaques patterns, respectively (Figure. S1). The GFPs of PMMoV-GFP2A are prominently presented on the surface of virions, forming fluorescent plaques in cells, thereby contributing to the observation of fluorescent plaques in systemic infection leaves. This information has been incorporated into the discussion section.

Reviewer 3 Report

Comments and Suggestions for Authors

The manuscript by Jiao and collaborators describes some green fluorescent protein (GFP)-labelling strategies to the coat protein of pepper mild mottle virus (PMMoV; tobamovirus) that allowed to make some interesting observation about CP accumulation dynamics during infection. Additionally, information about biological viability of labelled viral nanoparticles is also useful. The work was well planned and performed with the proper controls, and the manuscript has been well written and it is easy to understand.

In principle, I support publication of this manuscript, although I would like to encourage authors to improve the manuscript in two critical aspects. First, authors must enlarge and improve Discussion. In the current version of the manuscript, Discussion is mostly a brief repetition of some concepts already mentioned in Introduction. Second, although the Materials and Methods section can be easily followed, I strongly suggest that authors gather the whole sequences of all their viral clones in a Supplementary file. This will help readers to better understand constructs at the nucleotide level and will improve influence of this work into future tobamoviral research.

Finally, there are some minor details to correct. By the way, manuscript will be benefited of page and line numbering.

Page 2, 2nd paragraph, replace the major with a major viral pathogen

Page 2, 3rd paragraph, replace the word cleaved, since this is a bond skipping mechanism

Page 2, 2nd last sentence, what virus was 2A specifically derived?

Page 3, section 2.1 in Results, reformat the sentence to better express than A. tumefaciens was transformed with a vector in contrast to a vector was transformed into A. tumefaciens. In this second case, you could write that the vector was electroporated into (if electroporated). Attend this comment also in 2nd paragraph of section 2.4.

Author Response

Authors’ response to comments received

We thank the editor and the reviewers for their constructive comments and have now modified the manuscript to correct the errors and to respond to the comments and suggestions. In the text below, the comments received are in black and our replies and revisions in red.

Reviewer 3:

The manuscript by Jiao and collaborators describes some green fluorescent protein (GFP)-labelling strategies to the coat protein of pepper mild mottle virus (PMMoV; tobamovirus) that allowed to make some interesting observation about CP accumulation dynamics during infection. Additionally, information about biological viability of labelled viral nanoparticles is also useful. The work was well planned and performed with the proper controls, and the manuscript has been well written and it is easy to understand.

In principle, I support publication of this manuscript, although I would like to encourage authors to improve the manuscript in two critical aspects. First, authors must enlarge and improve Discussion. In the current version of the manuscript, Discussion is mostly a brief repetition of some concepts already mentioned in Introduction. Second, although the Materials and Methods section can be easily followed, I strongly suggest that authors gather the whole sequences of all their viral clones in a Supplementary file. This will help readers to better understand constructs at the nucleotide level and will improve influence of this work into future tobamoviral research.

Reply: We appreciate the insightful comments from the reviewer. Following your suggestions, we have revised the Materials and Methods section and included the sequences of viral clones in the Supplementary Materials.

Finally, there are some minor details to correct. By the way, manuscript will be benefited of page and line numbering.

Page 2, 2nd paragraph, replace the major with a major viral pathogen

 Reply: Corrected (Line 52).

Page 2, 3rd paragraph, replace the word cleaved, since this is a bond skipping mechanism

  Reply: Corrected (Line 85).

Page 2, 2nd last sentence, what virus was 2A specifically derived?

  Reply: Corrected (Line 84).

Page 3, section 2.1 in Results, reformat the sentence to better express than A. tumefaciens was transformed with a vector in contrast to a vector was transformed into A. tumefaciens.

Reply: Corrected (Lines 118-119).

 In this second case, you could write that the vector was electroporated into (if electroporated). Attend this comment also in 2nd paragraph of section 2.4.

 Reply: Corrected.

Round 2

Reviewer 1 Report

Comments and Suggestions for Authors

 Accept in present form

Reviewer 2 Report

Comments and Suggestions for Authors

The authors have provided the requested amendments to the ms. improving its general quality. The ms. can be published in its current form.